# The Effect of the Type of Activator Anion on the Hydration of Ground Granulated Blast Furnace Slag

**DOI:** 10.3390/ma15082835

**Published:** 2022-04-12

**Authors:** Łukasz P. Gołek, Wojciech Szudek, Michał Malik

**Affiliations:** 1Department of Building Materials Technology, Faculty of Materials Science and Ceramics, AGH University of Science and Technology, al. A. Mickiewicza 30, 30-059 Kraków, Poland; szudek@agh.edu.pl; 2Labotest Sp. z o.o. Laboratorium Inżynierii Lądowej, ul. Lwowska 38, 40-397 Katowice, Poland; michalmalik950426@gmail.com

**Keywords:** alkali activation, anions, alkali-activated materials, blast furnace slag

## Abstract

In this study, ground granulated blast furnace slag was activated with a wide variety of sodium salts to compare the effects of their pH and anion size on the hydration progress and compressive strength development of GGBFS pastes. Research was carried out on samples activated with twelve different sodium salts and cured for one year. Changes in their phase composition (XRD), loss on ignition at different temperatures, expansion and microstructure (SEM + EDS) were examined over the entire curing period. The results showed that the presence of sodium ions is more important than the pH of the system, as activation took place even in the case of compounds whose solutions are characterized by a low pH, such as sodium tartrate or phosphate. The compressive strength of the pastes ranged from approximately 8 to 65 MPa after one year of curing.

## 1. Introduction

Blast furnace slag is a by-product produced from the manufacturing of crude iron. Due to the recent changes in smelting technology, its market availability is gradually decreasing [1]. Ground granulated blast furnace slags have been traditionally used in the production of concretes and binders, along with other slags and waste materials from various metal processing branches and different mineral additives of industrial origin [2,3,4,5,6,7,8,9,10,11,12,13,14,15,16,17,18,19,20]. Slags with a high crystalline phase content, such as shaft slags, are used mainly as aggregates in road construction. On the other hand, copper and phosphorus slags can be used as components of alkali-activated binders [21]. According to prior research [22], granulated copper slag can be successfully activated, and its use in construction materials does not pose an environmental or health threat. Moreover, Łowińska-Kluge determined that the addition of copper slag can increase the resistance of cementitious materials to leaching, carbonation and acid attack by up to approx. 30% [23,24].

As a by-product, blast furnace slag has a low carbon footprint [4,25,26,27,28], unlike ordinary Portland cement that is manufactured in an energy intensive process, releasing an average of 790 kg of carbon dioxide during the production of 1 tonne of OPC (2017 data) [29], out of which 510 kg is associated with the decarbonation of CaCO_3_ alone [22]. Therefore, the use of slag allows for a reduction in the overall carbon footprint of cementitious composites, making them more sustainable. The sustainability aspect is important not only from an environmental point of view. All member states of the European Union are supposed to cut their greenhouse gas emissions to at least 55% below 1990 levels by 2030 and as a result, the trading prices of CO_2_ emission allowances are rapidly increasing, which forces the cement industry to seek alternatives to Portland clinker. This approach is reflected in the EN 197-1 standard, which specifies the content of additives in cement. In the case of blast furnace cement (CEM III/C), the content of slag can reach up to 95% [30]. However, the use of the material is supported by more than economic and environmental arguments.

Another important factor contributing to the popularity of ground granulated blast furnace slags is their positive impact on the properties of hydrates formed during binder setting and hardening. Research has shown that, compared to Portland cement, the C-S-H phase formed during the hydration of slag is characterized by a more compact microstructure. Therefore, slag binders are commonly used in concrete constructions that require a high resistance to corrosive environments [23]. Slag, as a material with latent hydraulic properties, must be activated in order to set and harden within a reasonable time frame. In cement chemistry, this process has been known for decades as alkali activation. It involves adding an additional raw material to either dry components or mixing water, which will increase the slag hydration rate by releasing excess alkali and OH^−^ ions into the system, thus increasing the pH of the paste and accelerating the dissolution of the silicate glass network [31]. The cheapest activators are sodium compounds. However, a very important but often overlooked aspect is the size of the anion accompanying the sodium cation. Typically used activators include sodium hydroxide, carbonate and silicate due to their availability and price. Nevertheless, for scientific purposes, other sodium compounds were also investigated in the present study.

The goal of this study was to determine the effect of various sodium salt anions on the progress of ground granulated blast furnace slag hydration. Compounds whose solutions are characterized by varied pH values were used in the experiments. The amount of sodium cations introduced by the activator was constant for all of the prepared samples. It has been known for decades that the slag hydration rate increases with pH value. Over the years, alkali activation was studied by Khül, Glukhovsky, Shi, Małolepszy and Deja, among others [21,32,33]. However, according to the research conducted by Bellmann and Stark, low-pH activators can also be used in the process [34]. This allows one to determine the impact of varying concentrations of sodium cations in the solution on the amount of hydration products formed.

## 2. Materials and Methods

Twelve analytical grade sodium compounds were used to activate the granulated blast furnace slag: benzoate (C_6_H_6_COONa), carbonate (Na_2_CO_3_), hypophosphite (NaH_2_PO_2_), tartrate (C_4_H_4_Na_2_O_6_·2H_2_O), citrate (C_6_H_6_O_7_Na_3_·2H_2_O), nitrate (NaNO_3_), phosphate (Na_3_PO_4_·12H_2_O), formate (CHNaO_2_), acetate (C_2_H_3_NaO_2_·3H_2_O), hydroxide (NaOH), thiosulfate (Na_2_O_3_S_2_) and chloride (NaCl). The compressive strength development of the pastes was examined over a period of one year. Changes in the linear dimensions of the samples, associated with hydrate formation and water binding, were also evaluated. Phase composition analysis (XRD) and SEM observations were carried out. Loss on ignition was measured during heating up to 800 °C.

### 2.1. Water/Slag Ratio

The appropriate amount of mixing water was determined experimentally based on slump flow measurements, the results of which are presented in Table 1 and Figure 1.

A W/S ratio of 0.28 was considered the most favorable and was used during sample preparation throughout the rest of the study.

### 2.2. Compressive Strength

For the compressive strength measurements, as the study was conducted only for comparative purposes, non-standard cubic samples with the dimensions of 20 mm × 20 mm × 20 mm were prepared. The amount of Na_2_O introduced with each of the sodium-containing activators was 5% by mass of slag. After being ground to a grain size below 63 µm in an agate mortar, the sodium compounds were added to the slag in solid form, and the dry components were homogenized. Then, distilled water was added and the pastes were mixed manually. The samples were formed into two portions, each compacted with a vibrating table. Molded pastes were placed over water in sealed polyethylene containers and cured at room temperature. For the purposes of the research, 216 slag-containing samples activated with 12 different sodium compounds were prepared. Their compressive strength was evaluated after 2, 7, 28, 90, 180 and 360 days of curing according to the EN 196-1 standard procedure, using a Controls Automax 5 universal testing machine (Milan, Italy). The values presented in the paper are the average of three measurements [35].

### 2.3. Changes in Linear Dimensions

Changes in linear dimensions were measured with a standard EN 12617-4:2002 [36], Graf-Kaufman apparatus on 20 mm × 20 mm × 160 mm prisms. The pastes were mixed according to the procedure described in Section 2.2. After 10 days, the samples were demolded, placed over water in sealed polyethylene containers and cured at room temperature. In the case of the samples containing sodium acetate, sodium citrate and sodium benzoate, no signs of binding and hardening were observed. Therefore, the pastes were prepared again and cured at 80 °C for 24 h in order to increase the hydration rate [35].

### 2.4. Loss on Ignition

Samples used to determine the loss on ignition during heating were prepared from the debris obtained after compressive strength measurements. The material was dried to a constant mass and ground to a grain size below 63 µm. Approx. 1 g of each sample was used for the study. The tests were carried out after 40 and 360 days of curing. The samples were heated up to 200, 400, 600 and 800 °C and kept at each temperature for 15 min. After each of the steps, they were weighed and their masses were compared.

### 2.5. Phase Composition

Samples analyzed by XRD were prepared according to the procedure described in Section 2.4. Approx. 2 g of each sample was used in the study. The diffractograms were collected after 40 and 360 days of curing, using a Philips PW 1050/70 diffractometer (Cu/Ni lamp operating at 35 kV/16 mA, 2θ range of 10–60°, step size of 0.05°, Amsterdam, the The Netherlands) and analyzed using X’Pert HighScore Plus software (PDF2 database, Malvern, UK).

### 2.6. SEM Observations

SEM observations were carried out on 16 fractures of the selected samples after 40 and 360 days of curing, using an FEI Nova NanoSEM 200 scanning electron microscope (Waltham, MA, USA).

## 3. Results

### 3.1. Compressive Strength

The results of the compressive strength measurements carried out after 2, 7, 28, 90, 180 and 360 days of curing are presented in Table 2 and Figure 2.

The highest compressive strength values were obtained for the samples activated with sodium carbonate. After one year of curing, a shrinkage-related decrease in strength was observed for the slag pastes activated with sodium phosphate, sodium benzoate, sodium tartrate, sodium citrate, sodium thiosulfate and sodium hydroxide. The strength development of samples containing Na_2_CO_3_ was superior, compared to other series; however, the effect is most likely related to carbonate formation, as was described in the third model proposed by Shi [37]. An important factor that might have affected the compressive strength of the pastes is the varying mass of the activators used, as in the case of some of the salts, maintaining the 5 wt% Na_2_O equivalent required the introduction of large amounts of compounds. The relative amount of activators, calculated with respect to the mass of sodium hydroxide, is presented in Figure 3.

As can be seen, over 3.5 times more sodium benzoate was introduced as an activator during sample preparation, compared to sodium hydroxide, which might have had an impact on the porosity of the paste, significantly decreasing its compressive strength. The highest strength values were obtained for the samples containing the lowest amounts of the activators. However, the factor that could be much more important than the slag “dilution effect” is the varying pH of the activator. This relationship is presented in Figure 4.

As expected, the samples containing the strongest basic activators were characterized by the highest compressive strength values and the most rapid strength development. However, the activation also took place at low pH levels, as evidenced by the increase in strength of all of the samples between 28 and 360 days of curing. This may indicate that the concentration of Na^+^ ions in the hydrating system was high enough to lead to the diffusion of Ca^2+^ ions from the slag, which affected the metastable structure of the glass, and, after a longer curing period, broke the Al-O and Si-O chemical bonds, allowing for a gradual hydration progress and leading to the formation of the C-S-H phase.

### 3.2. Changes in Linear Dimensions

The results of the elongation measurements, performed with a Graf-Kaufman apparatus, are presented in Table 3 and Figure 5.

The greatest expansion was observed in the case of the prisms activated with sodium tartrate, sodium nitrate and sodium phosphate. This effect may have resulted from the formation of amorphous silica gel, which absorbs water and increases its volume, leading to sample elongation [38]. Interesting results were obtained for slag activated with sodium tartrate: the expansion rate increased rapidly after 28 days of hydration, which correlates well with the considerable increase in the compressive strength observed between 28 and 90 days of curing.

### 3.3. Loss on Ignition

As is clearly visible in Figure 6, the LOI values increased considerably after a longer curing period. The effect is related to the dehydration of calcium (aluminum) silicate hydrates, present in greater amounts in the older pastes. The mass loss was observed for all of the samples, proving the progress of hydration, and also in the case of those activated with sodium salts whose solutions were characterized by a low pH. The dehydration of C-(A)-S-H and ettringite takes place in the temperature range of 20–440 °C [39]. In the range of 200–600 °C, the organic part of the activator decomposes, which was observed for pastes containing sodium tartrate, benzoate, citrate and acetate. In the case of these salts, the decomposition of anions and the combustion of the organic carbon took place. The temperature range of 440–570 °C covers the dehydroxylation of calcium hydroxide. Afterwards, with temperatures of up to 820 °C, the decarbonation of CaCO_3_ is observed [40]. The mass loss above 600 °C occurred mostly in the case of older samples, which proves that they were partially carbonated.

### 3.4. XRD Analysis

The diffractograms of the slags were compiled based on the type of activator used (low/high pH). After 40 days of hydration, the diffractograms of the pastes activated with sodium hydroxide and sodium phosphate (Figure 7) showed reflexes characteristic of C-S-H phase crystallites, which indicates their rapid hydration, and correlates well with other studies. However, after 360 days, the samples were partially carbonated. A similar outcome was observed for pastes activated with sodium carbonate, but in this case, calcium carbonate was present in the diffractogram even after 40 days of hydration.

Different results were obtained for the samples activated with sodium compounds characterized by a lower pH (Figure 8).

For all of the samples presented in Figure 8, after 40 days of curing, the intensity of the C-S-H phase characteristic peak was much lower than that of the pastes activated with high pH compounds. In the case of sodium formate and sodium tartrate, the diffractograms did not differ significantly from that of unreacted ground granulated blast furnace slag. After 360 days of curing, a significant increase in the amount of the C-S-H phase was observed for all samples, combined with their partial carbonation. Traces of unreacted activators—sodium benzoate, phosphate and nitrate—were also found in the appropriate pastes.

### 3.5. SEM Observations

SEM observations were carried out on selected samples, characterized by the highest (Figure 9, Figure 10, Figure 11, Figure 12, Figure 13 and Figure 14) and the lowest (Figure 15, Figure 16, Figure 17, Figure 18, Figure 19 and Figure 20) pH values of the activator solutions. In the case of the fracture activated with NaOH, after 40 days of hydration, finely crystalline C-S-H gel was visible, as proven by the EDS microanalysis. The phase, poorly bound to the surface of the slag grains on which it was formed, was characterized by a high sodium concentration in its structure. This changed after 360 days of hydration. Then, fractured slag particles were found, which proves the high mechanical strength and cohesion of the hydrates, exceeding that of unreacted grains. Furthermore, it could be seen that the C-S-H phase formed after one year of curing contained a high amount of silica.

In the case of the sample activated with Na_2_CO_3_, after 40 days of hydration, the outlines of slag grains covered with a layer of hydrates were observed. The EDS microanalysis of the fracture showed only an amorphous C-S-H phase. After one year, a much greater amount of the phase was visible, containing a high amount of sodium in its structure. Furthermore, semi-crystalline silica gel appeared.

The fractured slag grain found during the observations of the paste activated with sodium phosphate proves the strong bonding of hydrates with the surface of the glass, as early as after 40 days of curing. The EDS microanalysis allowed for a conclusion that the C-S-H phase formed on the surface of the grains contained phosphorus, originating from the activator anions. After one year, the slag grains were covered by a compact layer of hydrates.

The microstructures of the pastes containing low-pH sodium compounds are presented in Figure 15, Figure 16, Figure 17, Figure 18, Figure 19 and Figure 20. After 40 days of curing, an SEM analysis of the fractures of slag activated with sodium nitrate showed only a thin layer of hydrates covering the surface of the grains. The layer was characterized by a high C/S ratio, which might indicate that after such a short hydration period, the structure of the glass was still unaffected and, therefore, only a small amount of Si^4+^ ions had been leached into the solution, forming the C-S-H phase. After one year of hydration, the fractures of the same sample showed a considerable increase in the C-S-H content. However, compared to the EDS analyses of the samples activated with strongly basic compounds, the hydrates were characterized by a substantially higher calcium content. A similar effect was noticed in the case of the pastes activated with sodium formate; after a longer hydration period, the structure of the C-S-H phase became more compact, but calcium was still predominant in its chemical composition. Furthermore, it was characterized by a loose microstructure that exhibited considerable porosity. After 40 days of curing, in the early hydration stage, the grains of the slag activated with NaCl were covered with strongly bound, porous hydrates. Meanwhile, after one year, the fractures showed characteristic plate-like structures, containing a high amount of chlorine.

## 4. Conclusions

The analysis of the results allowed for an evaluation of the impact of the type of activator anion on the progress of slag hydration and the effectiveness of the activation itself. It was observed that alkali activation occurs not only in the case of compounds whose solutions are characterized by high pH values, e.g., sodium chloride, but also for neutral or even low-pH reagents. Lowering the pH of the activator and increasing the size of its anion both have a negative effect on the efficiency of the activation; however, the impact of anion size is more pronounced. Additionally, the examinations clearly show that the presence of sodium ions is more important than the pH of the activator, which should be taken into account when designing the composition of future alkali-activated binders.

Based on the obtained results, the following conclusions were drawn:Strongly basic compounds are the most effective activators of blast furnace slags.Strength development was observed for all of the samples prepared in the study, which proves that sodium activation occurs even at a low or neutral pH.Despite the relatively low compressive strength values obtained for pastes activated with low-pH compounds, the microstructure of hydrates formed in the case of, e.g., sodium chloride, was compact, and they were strongly bonded to the surface of the slag grains, which shows potential for further research.Slag activation with sodium carbonate provides high early compressive strength values after 7 days of curing and does not entail changes in the linear dimensions of the samples over time that could lead to the destruction of the material.Most of the samples cured for 360 days were carbonated.

## Figures and Tables

**Figure 1 materials-15-02835-f001:**
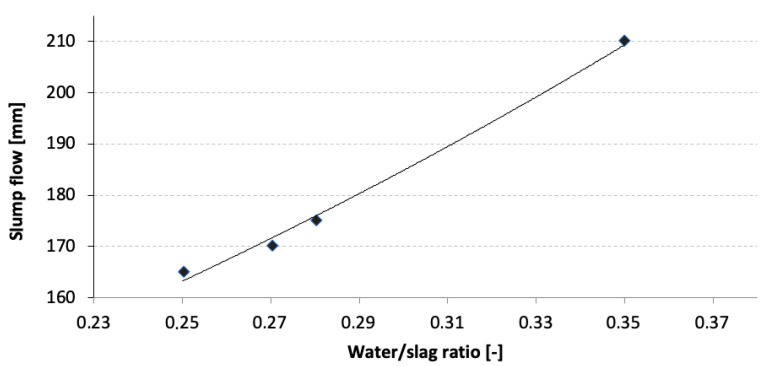
Paste flow as a function of water/slag ratio.

**Figure 2 materials-15-02835-f002:**
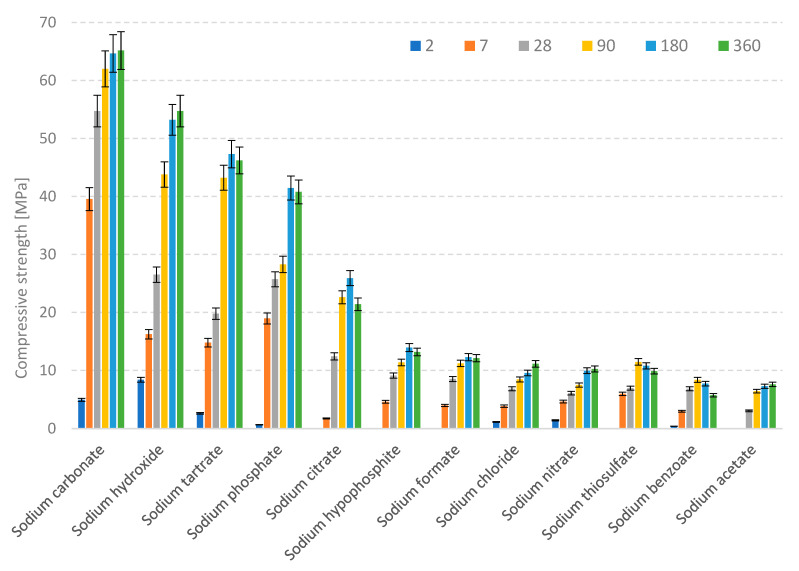
Changes in the compressive strength of pastes over one year of hydration.

**Figure 3 materials-15-02835-f003:**
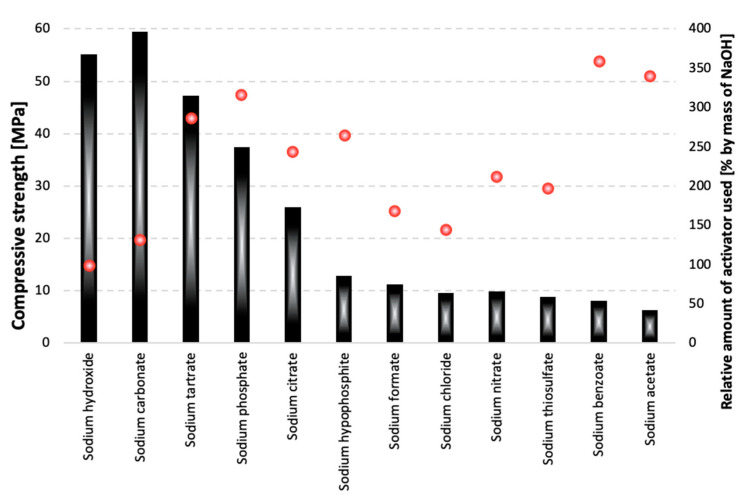
The relation between the compressive strength of pastes after 180 days of curing and the amount of activator used in their preparation. Red dots—relative amount of activator.

**Figure 4 materials-15-02835-f004:**
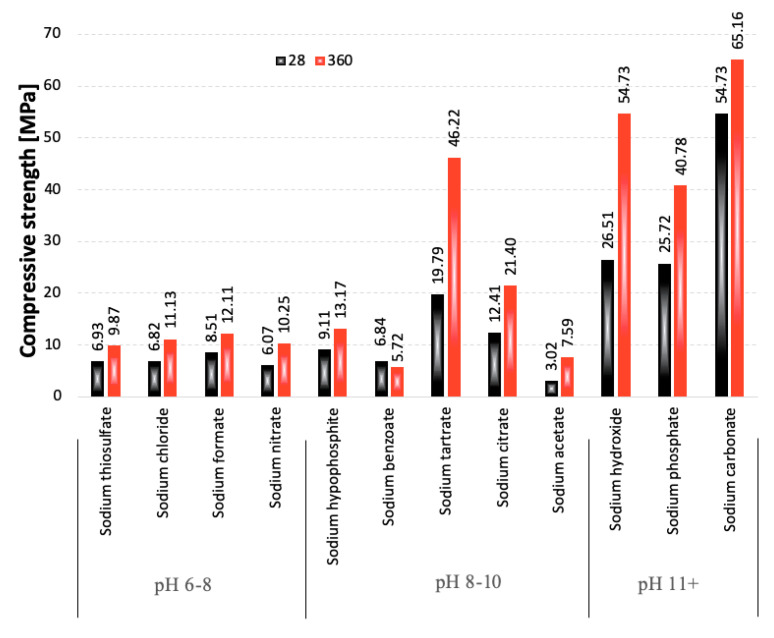
Compressive strength of the pastes after 28 days of curing, ranked according to the pH of the activator solutions.

**Figure 5 materials-15-02835-f005:**
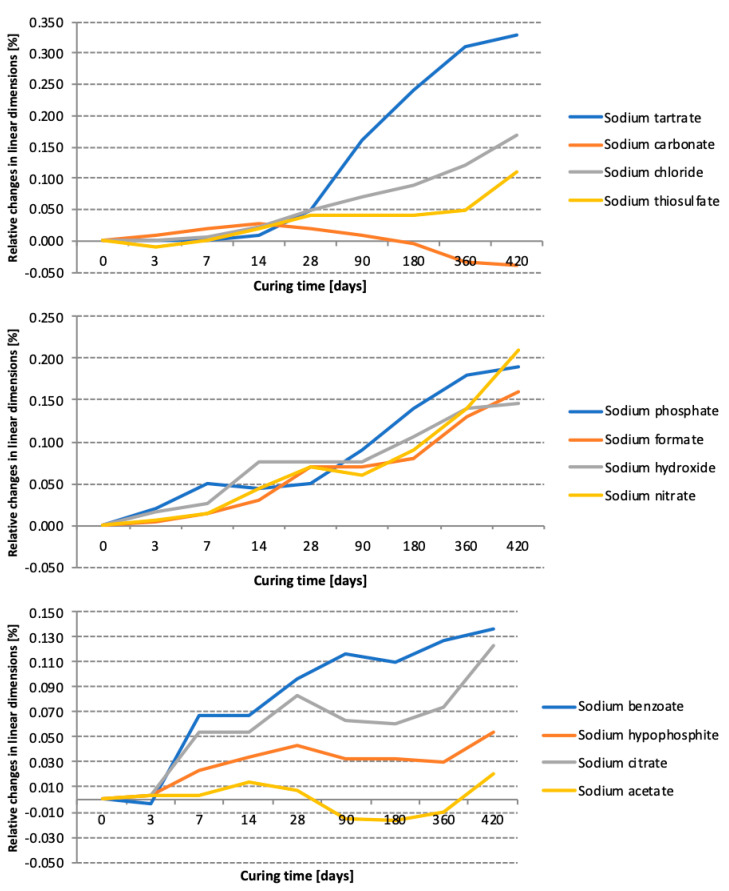
Changes in the linear dimensions of paste prisms depending on the type of activator used.

**Figure 6 materials-15-02835-f006:**
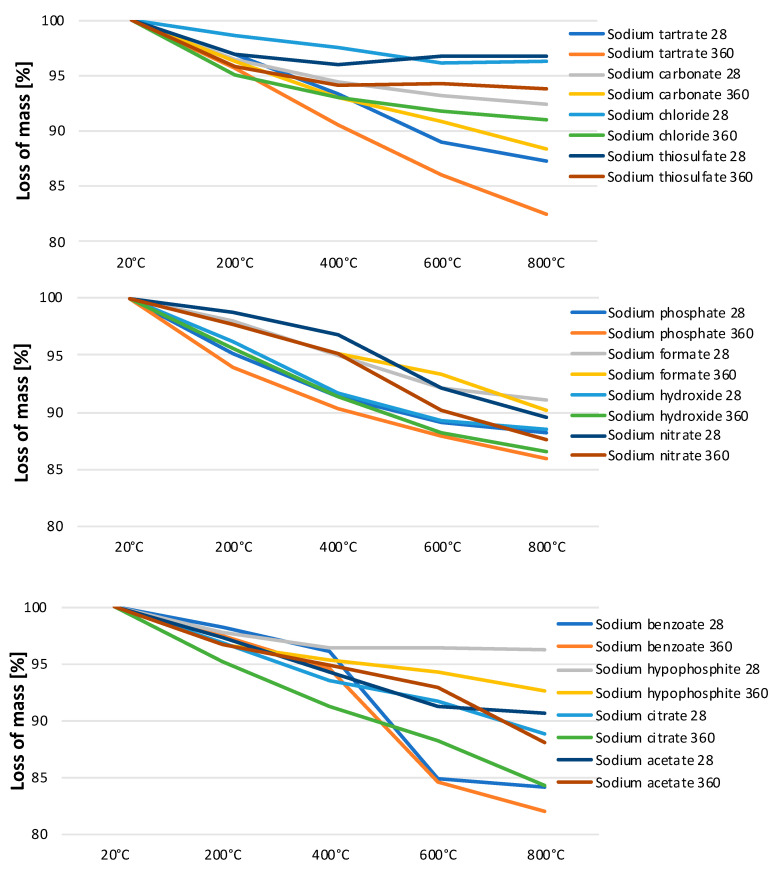
Loss on ignition during heating.

**Figure 7 materials-15-02835-f007:**
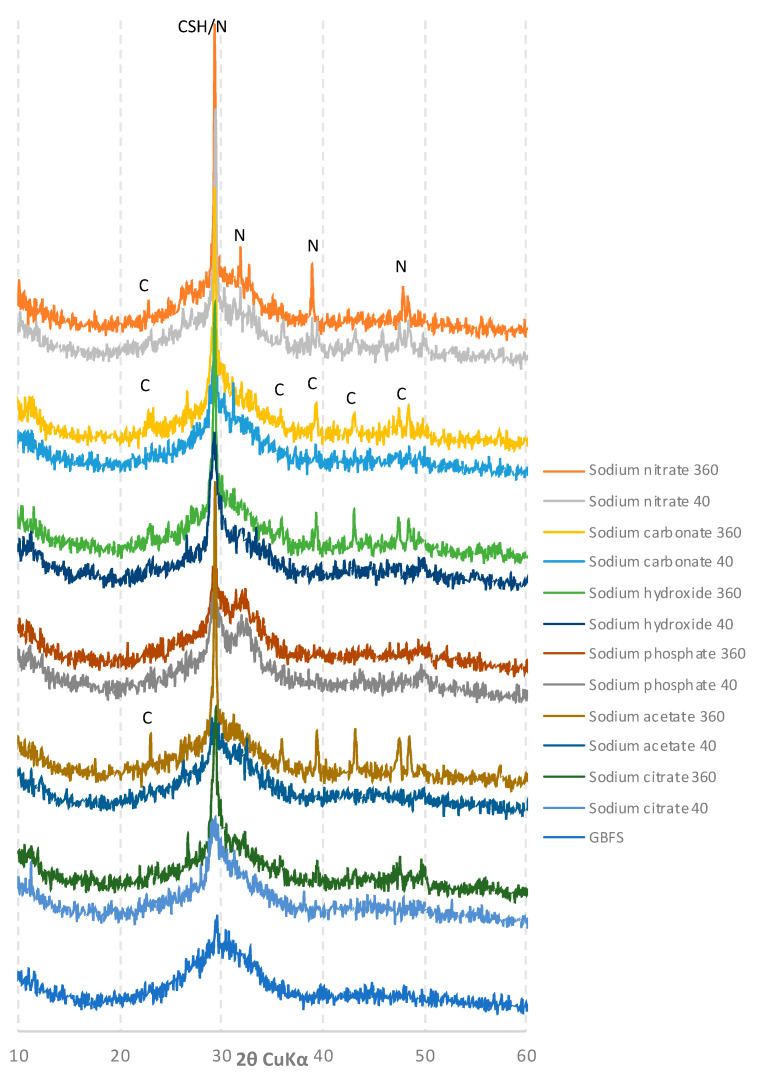
Diffractograms of slag activated with sodium nitrate, carbonate, hydroxide, phosphate, acetate and citrate after 40 and 360 days of curing: **CSH**—calcium silicate hydrates; **N**—sodium nitrate; **C**—calcite.

**Figure 8 materials-15-02835-f008:**
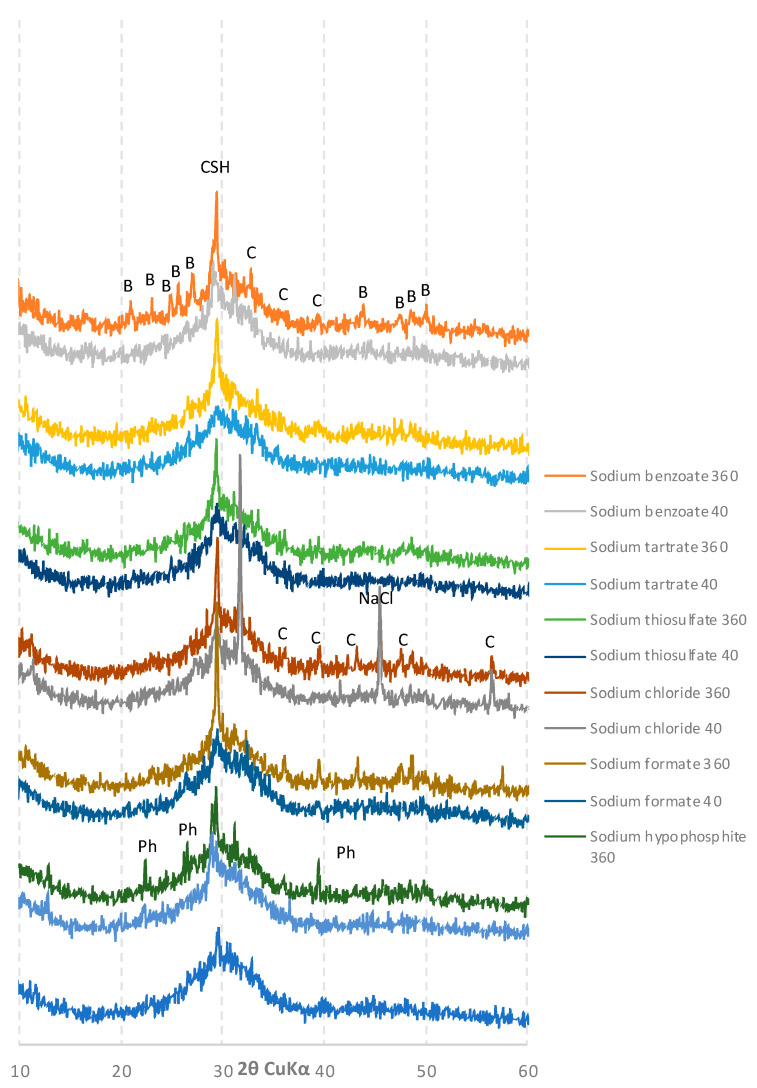
Diffractograms of slag activated with sodium benzoate, tartrate, thiosulfate, formate, chloride and hypophosphite after 40 and 360 days of curing: **CSH**—calcium silicate hydrates; **B**—sodium benzoate; **C**—calcite; **NaCl**—sodium chloride; **Ph**—sodium phosphate.

**Figure 9 materials-15-02835-f009:**
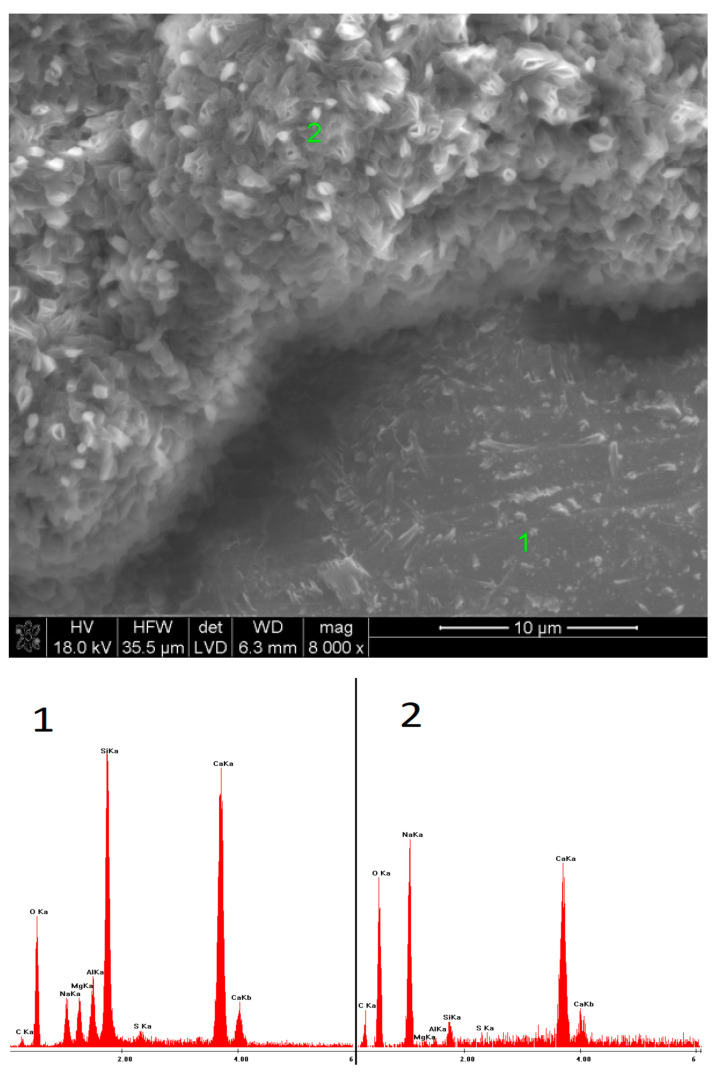
Microstructure of slag paste activated with sodium hydroxide after 40 days of curing—SEM + EDS; **1**—hydration products; **2**—unreacted slag grain.

**Figure 10 materials-15-02835-f010:**
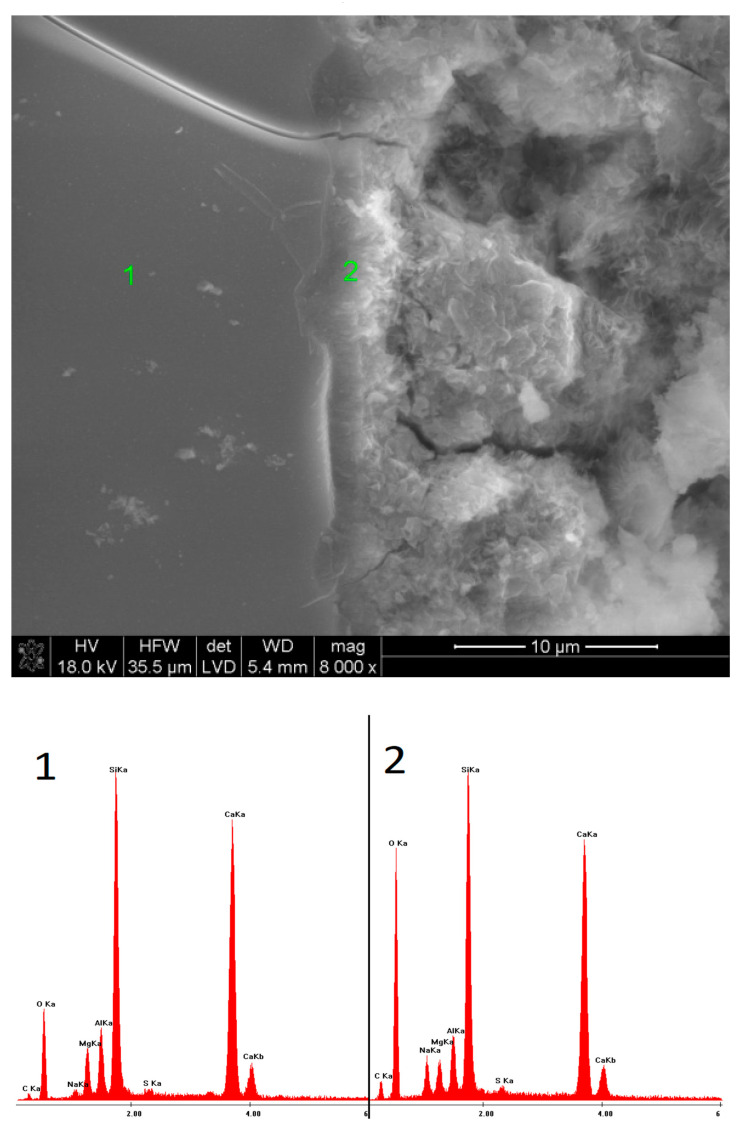
Microstructure of slag paste activated with sodium hydroxide after 360 days of curing—SEM + EDS; **1**—unreacted slag grain; **2**—hydration products.

**Figure 11 materials-15-02835-f011:**
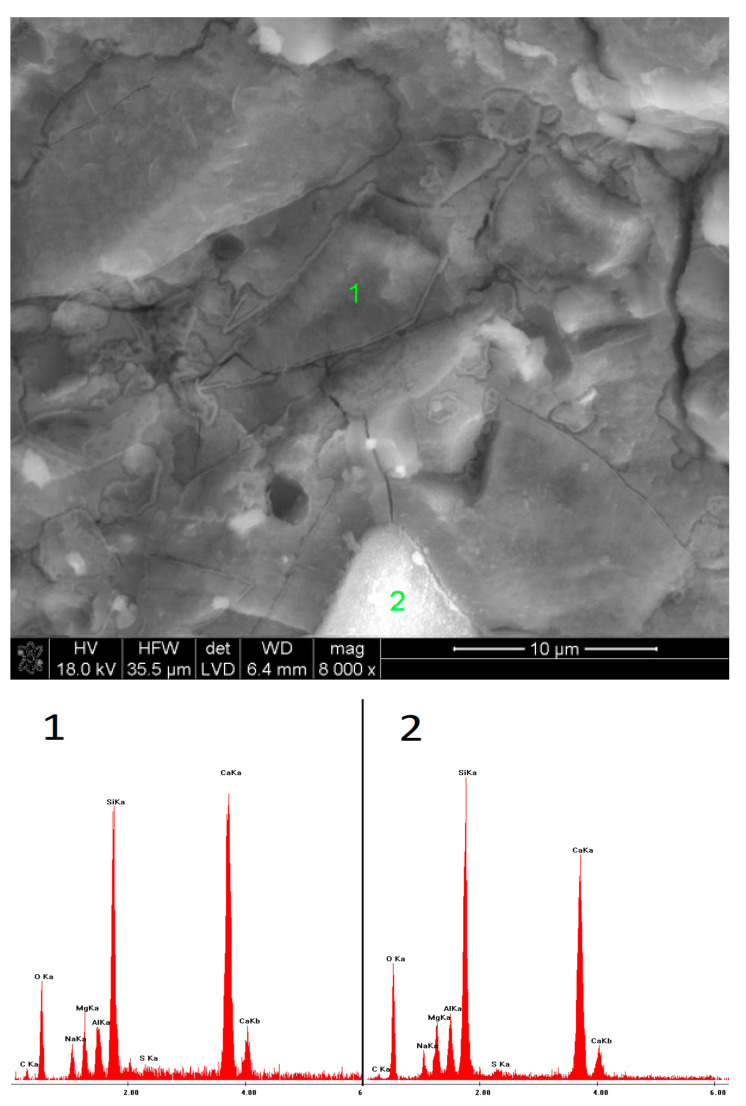
Microstructure of slag paste activated with sodium carbonate after 40 days of curing—SEM + EDS; **1**,**2**—slag grains covered with hydration products.

**Figure 12 materials-15-02835-f012:**
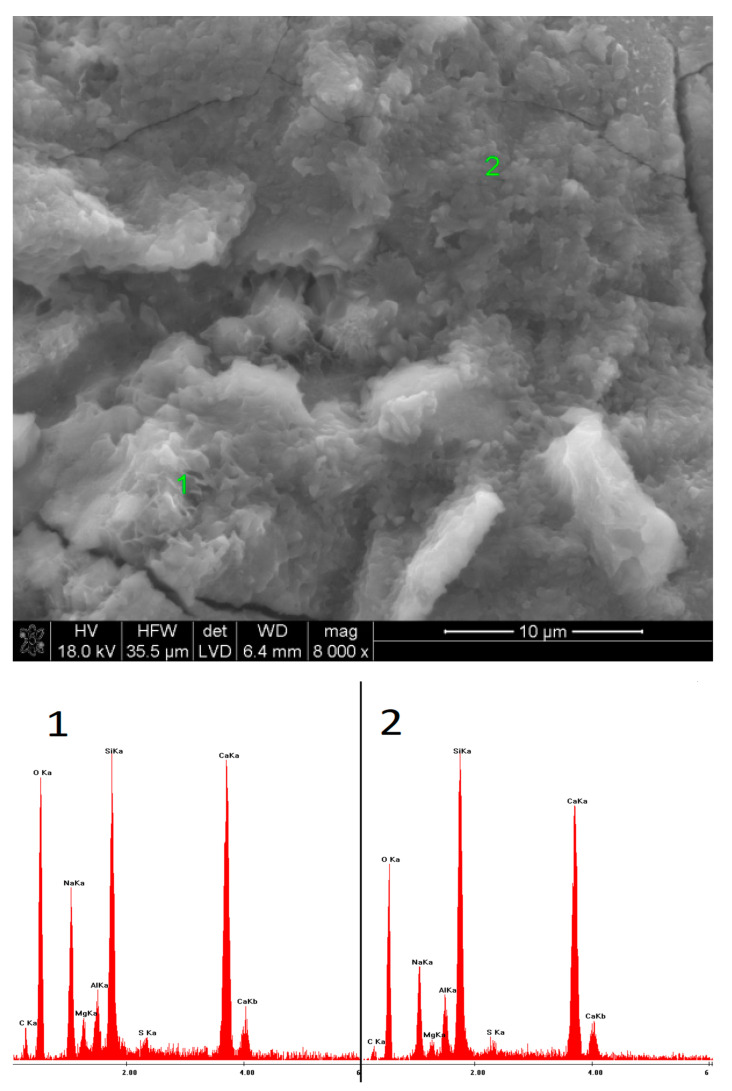
Microstructure of slag paste activated with sodium carbonate after 360 days of curing—SEM + EDS; **1**,**2**—hydration products.

**Figure 13 materials-15-02835-f013:**
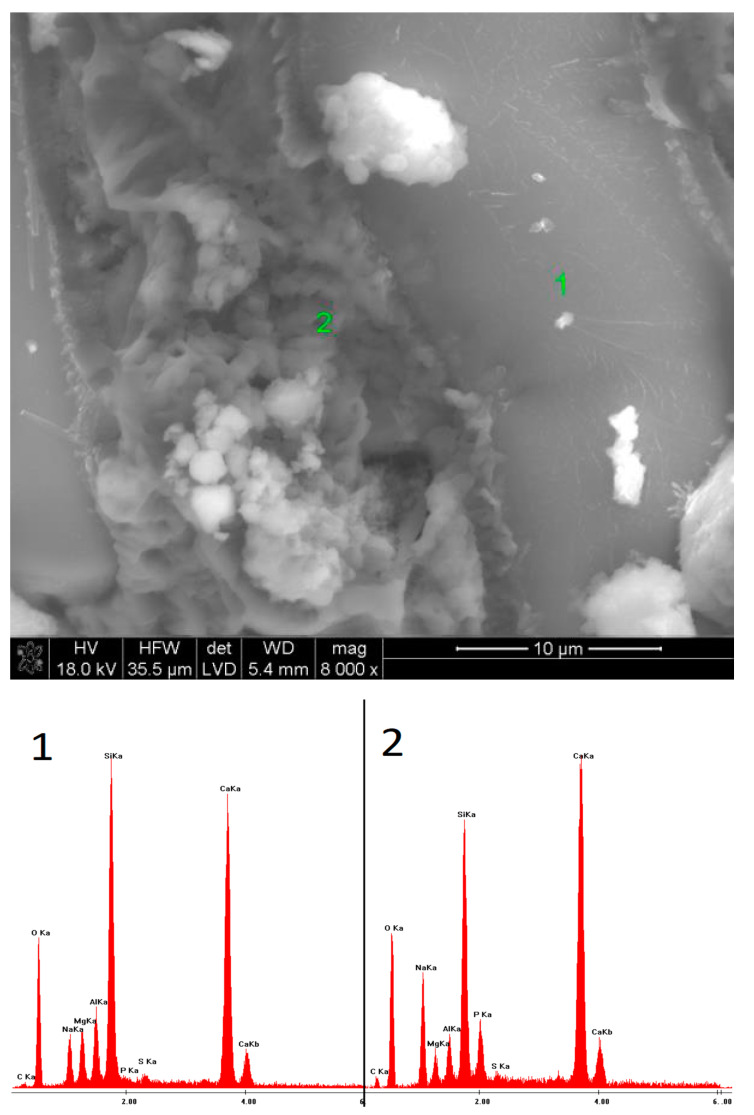
Microstructure of slag paste activated with sodium hypophosphite after 40 days of curing—SEM + EDS; **1**—unreacted slag grain; **2**—hydration products.

**Figure 14 materials-15-02835-f014:**
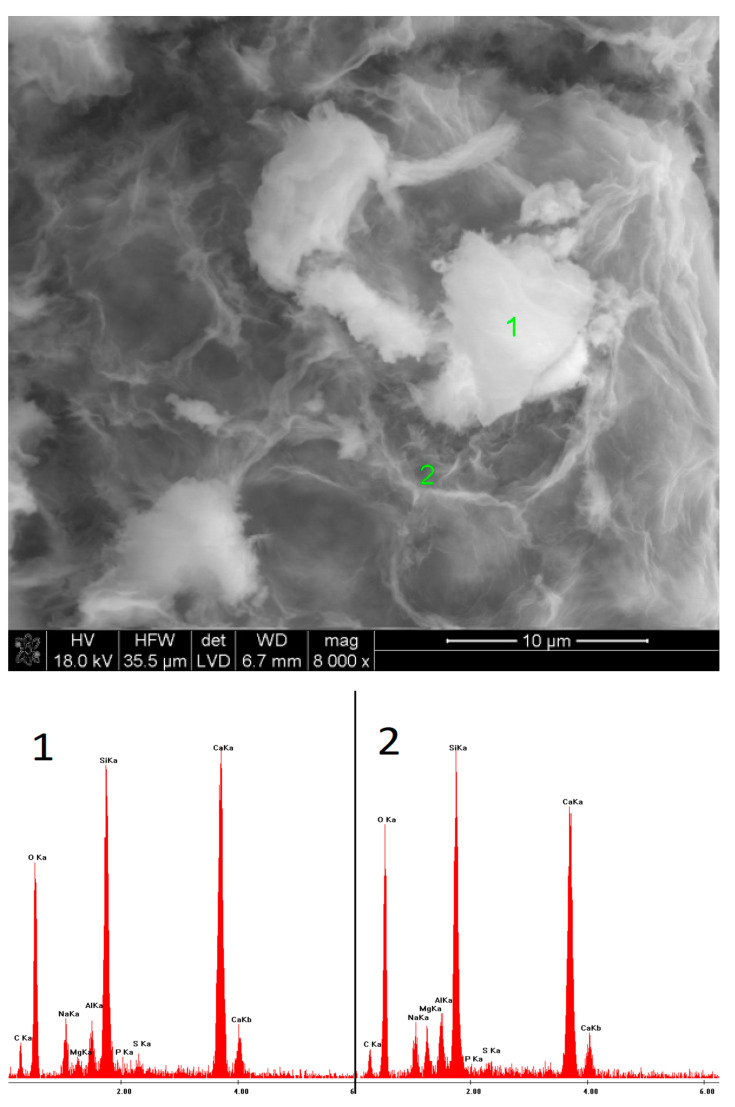
Microstructure of slag paste activated with sodium hypophosphite after 360 days of curing—SEM + EDS; **1**,**2**—a layer of hydration products covering the slag grains.

**Figure 15 materials-15-02835-f015:**
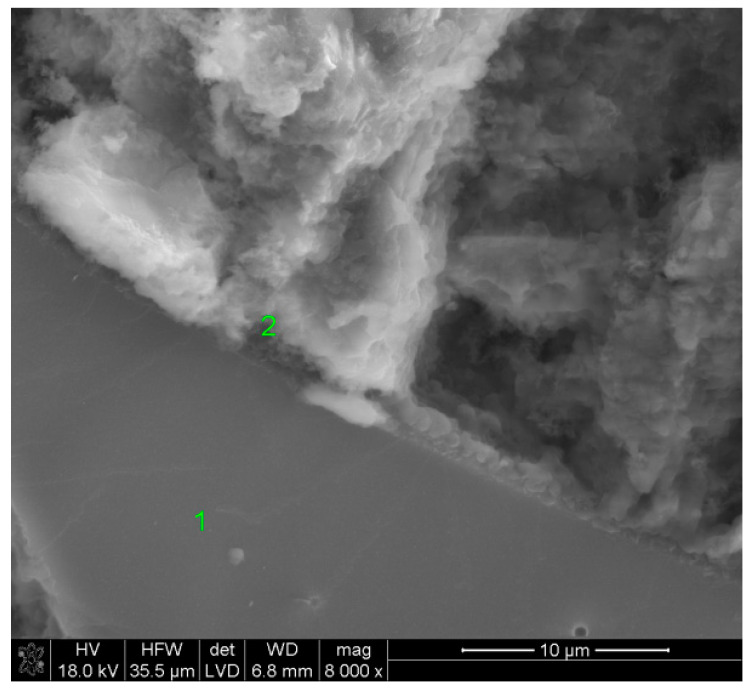
Microstructure of slag paste activated with sodium nitrate after 40 days of curing—SEM + EDS; **1**—unreacted slag grains; **2**—hydration products.

**Figure 16 materials-15-02835-f016:**
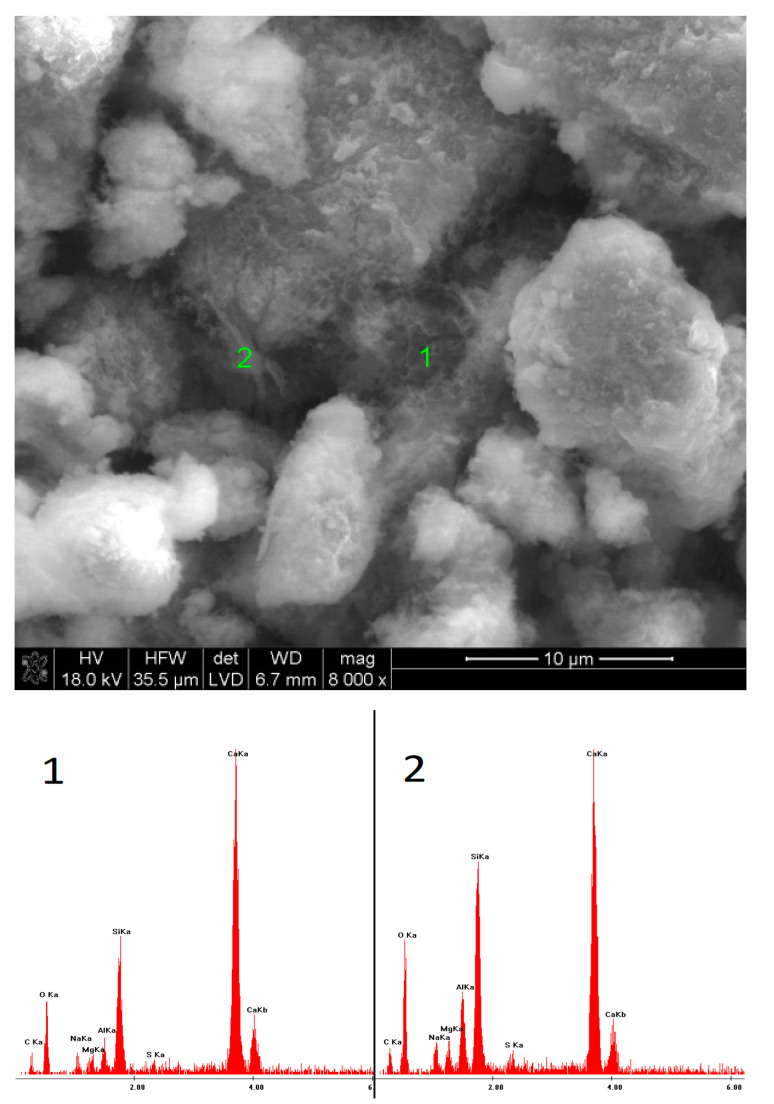
Microstructure of slag paste activated with sodium nitrate after 360 days of curing—SEM + EDS; **1**,**2**—hydration products covering the slag grains.

**Figure 17 materials-15-02835-f017:**
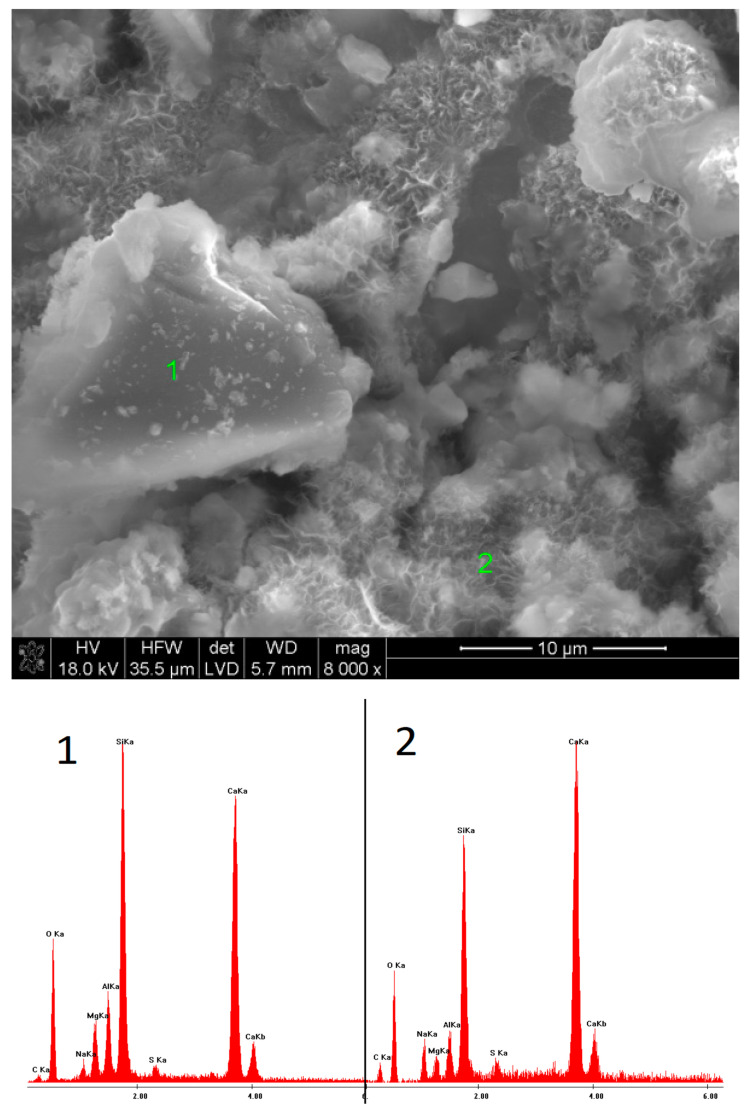
Microstructure of slag paste activated with sodium formate after 40 days of curing—SEM + EDS; **1**—slag grain; **2**—hydration products.

**Figure 18 materials-15-02835-f018:**
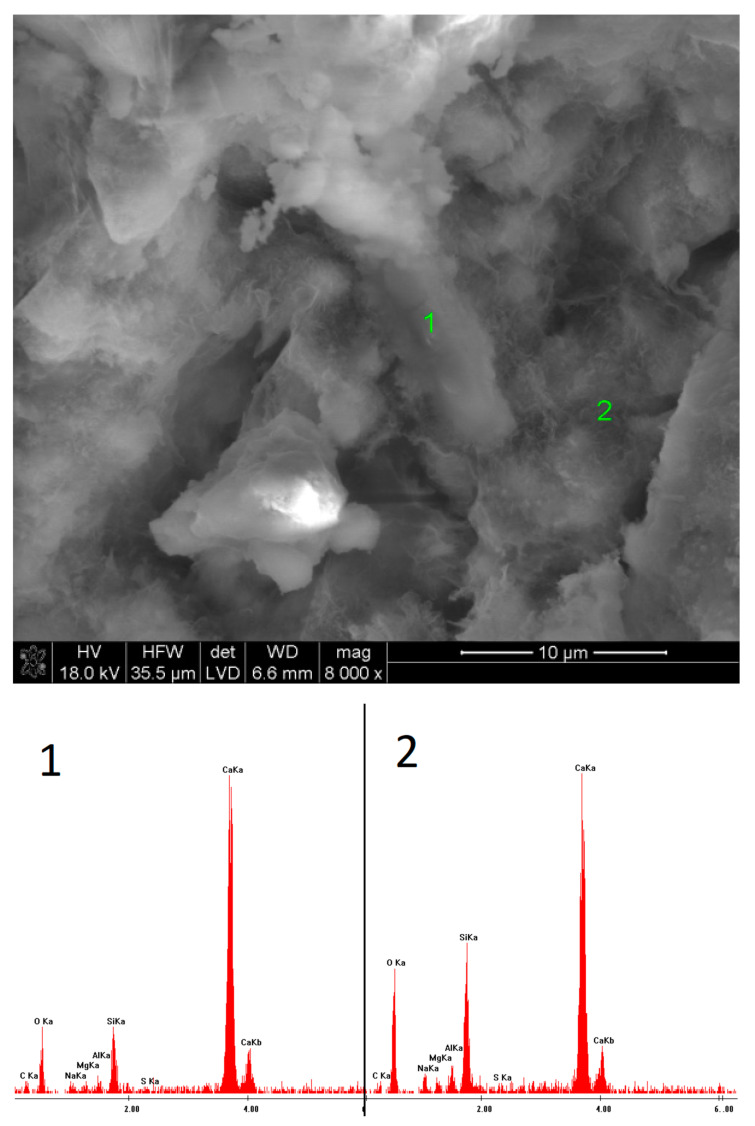
Microstructure of slag paste activated with sodium formate after 360 days of curing—SEM + EDS; **1**,**2**—hydration products covering the slag grains.

**Figure 19 materials-15-02835-f019:**
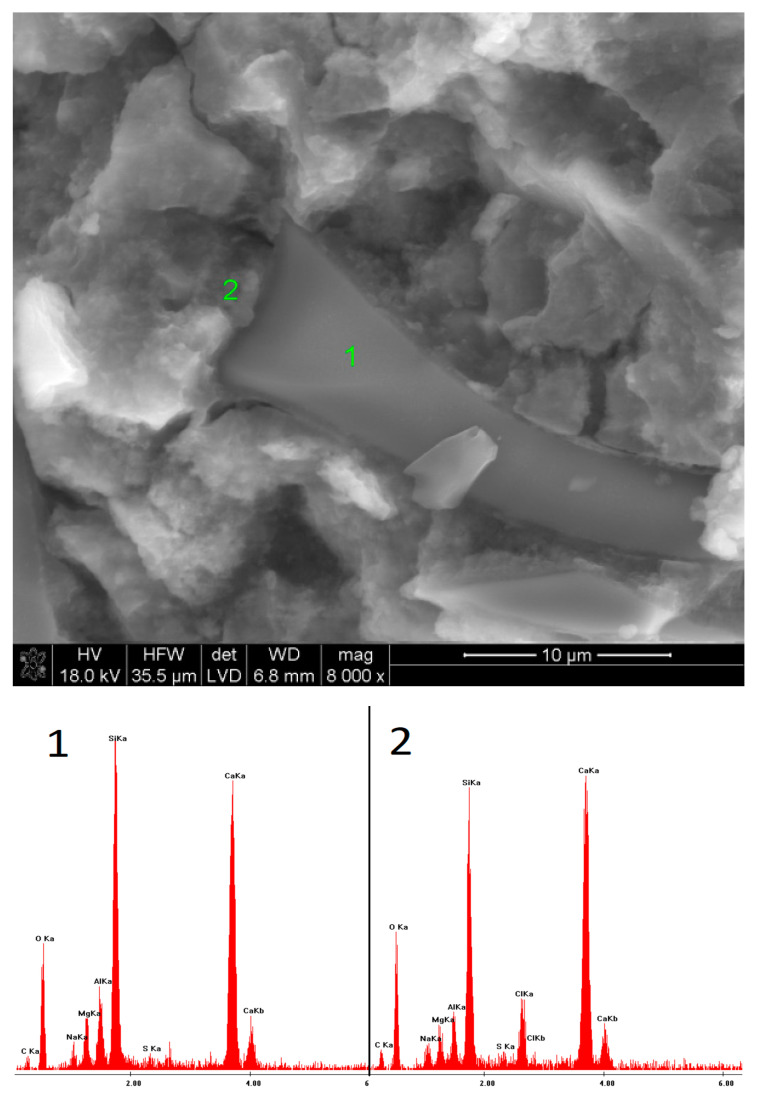
Microstructure of slag paste activated with sodium chloride after 40 days of curing—SEM + EDS; **1**—unreacted slag grain; **2**—hydration products.

**Figure 20 materials-15-02835-f020:**
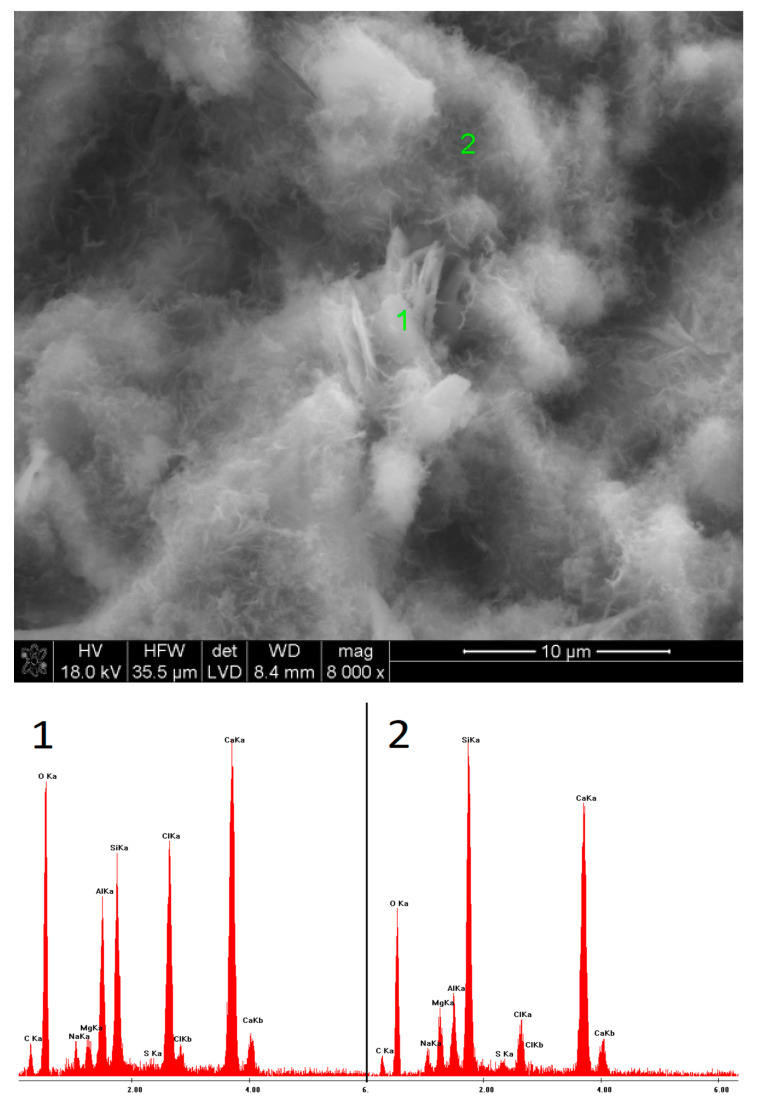
Microstructure of slag paste activated with sodium chloride after 360 days of curing—SEM + EDS; **1**,**2**—hydration products.

**Table 1 materials-15-02835-t001:** Paste flow at different water/slag ratios.

W/S Ratio [-]	Flow [mm]
0.25	165
0.27	170
0.28	175
0.35	210

**Table 2 materials-15-02835-t002:** Compressive strength of pastes activated with different sodium compounds.

	Compressive Strength [MPa]					
Activator	2 Days	7 Days	28 Days	90 Days	180 Days	360 Days
Sodium benzoate (C_6_H_6_COONa)	0.41	2.96	7.74	8.37	8.86	5.72
Sodium carbonate (Na_2_CO_3_)	4.92	39.55	34.57	68.37	59.45	53.03
Sodium hypophosphite (NaH_2_PO_2_)	0.00	4.59	9.11	14.38	12.94	16.97
Sodium tartrate (C_4_H_4_Na_2_O_6_·2H_2_O)	2.59	14.78	16.79	44.07	47.31	32.02
Sodium citrate (C_6_H_6_O_7_Na_3_·2H_2_O)	0.00	1.72	8.41	20.28	26.43	19.31
Sodium nitrate (NaNO_3_)	1.41	4.63	6.07	7.47	9.94	10.25
Sodium phosphate (Na_3_PO_4_·12H_2_O)	0.66	18.96	30.86	35.73	44.24	44.22
Sodium formate (CHNaO_2_)	0.00	3.95	8.51	12.21	11.28	13.10
Sodium acetate (C_2_H_3_NaO_2_·3H_2_O)	0.00	0.00	3.02	6.42	6.28	6.59
Sodium hydroxide (NaOH)	8.39	16.22	26.51	43.78	55.12	49.02
Sodium thiosulfate (Na_2_O_3_S_2_)	0.00	5.96	6.93	11.46	8.78	9.87
Sodium chloride (NaCl)	1.15	3.83	6.82	8.43	9.56	11.13

**Table 3 materials-15-02835-t003:** Changes in the linear dimensions of paste prisms.

	Elongation [mm]							
	0 Days	3 Days	7 Days	14 Days	28 Days	90 Days	360 Days	420 Days
Sodium tartrate (C_4_H_4_Na_2_O_6_·2H_2_O)	0.000	0.000	0.000	0.010	0.050	0.160	0.240	0.310
Sodium carbonate (Na_2_CO_3_)	0.000	0.010	0.020	0.027	0.020	0.010	−0.005	−0.035
Sodium chloride (NaCl)	0.000	0.000	0.007	0.023	0.050	0.070	0.090	0.120
Sodium thiosulfate (Na_2_O_3_S_2_)	0.000	−0.010	0.000	0.020	0.040	0.040	0.040	0.050
Sodium phosphate (Na_3_PO_4_·12H_2_O)	0.000	0.020	0.050	0.043	0.050	0.090	0.140	0.180
Sodium formate (CHNaO_2_)	0.000	0.003	0.013	0.030	0.070	0.070	0.080	0.130
Sodium hydroxide (NaOH)	0.000	0.017	0.027	0.077	0.077	0.077	0.107	0.140
Sodium nitrate (NaNO_3_)	0.000	0.007	0.013	0.045	0.070	0.060	0.090	0.140
Sodium benzoate (C_6_H_6_COONa)	0.000	−0.003	0.067	0.067	0.097	0.117	0.110	0.127
Sodium hypophosphite (NaH_2_PO_2_)	0.000	0.003	0.023	0.033	0.043	0.033	0.033	0.030
Sodium citrate (C_6_H_6_O_7_Na_3_·2H_2_O)	0.000	0.003	0.053	0.053	0.083	0.063	0.060	0.073
Sodium acetate (C_2_H_3_NaO_2_·3H_2_O)	0.000	0.003	0.003	0.013	0.007	−0.015	−0.017	−0.010

## Data Availability

The data presented in this study are available upon request from the corresponding author.

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
