# Peer review of "The Effect of the Type of Activator Anion on the Hydration of Ground Granulated Blast Furnace Slag"

_materials, 2022, doi:10.3390/ma15082835_

Round 1

Reviewer 1 Report

Pls see the attached file

Author Response

I am very grateful for your time. I appreciate your help and your tips are important to me. All the responses are included in the attached file.

Reviewer 2 Report

The article “The effect of the type of activator anion on the hydration of ground granulated blast furnace slag” presents an interesting topic. However, the following minor revision is suggested before the consideration for publication.

  1. Line number 39-43, The statement requires a reference.
  2. Line number 71-73, please revise to understand clearly.
  3. Line number 107, Changes in linear dimensions were measured with a standard EN 1367-4 Graf-Kaufman apparatus on 20 x 20 x 160 mm prisms, give a reference for EN 1367-4
  4. Please add the method of curing to the materials and method section.
  5. On line 225, After 360 days of curing, a significant increase in the amount of the C-S-H phase was observed for all samples, please explain why?
  6. Also, please mention if there is any phase difference in XRD based on curing age.
  7. On line 241-242, After one year, a much greater amount of the phase was visible, containing a high amount of sodium in its structure, please explain a reason for that.
  8. Improve the conclusion section of the study as it is poorly written.
  9. To support the background of the study, the introduction section should be improved with more recent references.

Author Response

(The authors gave the same response as above.)

Reviewer 3 Report

Gołek and coworkers presented a very interesting work about the effect of the type of activator anion on the hydration of ground granulated blast furnace slag. The experiments are performed systematically, and the paper is well written. Therefore, I suggest publishing this paper with the following revisions.

  1. The figure quality must be improved. These includes a. Figure 7, the title should not on the y axis; b. Figure 8, the color of different lines is pretty close. The color must be revised so it is easier to tell the differences. c. The unit of y axis in the EDS figure should be given.
  2. The discussion about SEM and EDS is not very easy to follow. I think the authors should revise this part, showing what is the difference of 1 and 2 region and what it suggests. Is there a phase separation happened?
  3. The authors should double-check the typos in the whole paper. For example, page 6, “„dilution effect””.
  4. The enhanced compressive strength is interesting. The highest compressive strength values were obtained for the samples activated with sodium carbonate. The authors said it is because of the formation of CaCO3. Is the cation, such as Na, also plays a role?
  5. I think some more discussion about the enhanced mechanical properties could be added. The authors have a lot of data on different salts and their compressive strength, and some discussion about the structures of the anion and their performance is useful for readers.
  6. Following the question 5, forming compositions to enhance the mechanical properties is also well known in carbon nanotube doped composites. Some papers (https://doi.org/10.1039/D1PY00705J; Adv. Mater., 2006, 18 , 689 —706) about carbon nanotube enhance the composites mechanical properties could be cited somewhere appropriate.

Author Response

(The authors gave the same response as above.)

Reviewer 4 Report

In this paper the effect of the type of activator anion on the hydration of ground granulated blast furnace slag was investigated using different analysis, tests and measurements such as compressive strength measurements, changes in linear dimensions, loss on ignition, XRD and SEM analysis.

From my point of view there are some aspects to improve:

1.The References should be improved. You can cite the following paper:

Sabău, E.; Udroiu, R.; Bere, P.; Buranský, I.; Miron-Borzan, C.-Ş. A Novel Polymer Concrete Composite with GFRP Waste: Applications, Morphology, and Porosity Characterization. Appl. Sci. 2020, 10, 2060. https://doi.org/10.3390/app10062060

2.How the compressive strength tests were performed? What machine was used? Also, please specify the standards used.

3.All the devices used to test and analyze the samples should be mentioned in the Materials and Methods section. This information is missing.

4.The Figures 9 to 20 are too large. The size of them should be reduced. Also, these 11 Figures should be detail described.

5. The abbreviations should be described at the first used.

6.What is new in this research? Please mention it in the conclusion section.

7.Please mention the applications of this study.

8.Are the limitations of this study noted? The limitations of this study should be discussed.

Author Response

(The authors gave the same response as above.)

Round 2

Reviewer 4 Report

All the comments are addressed well and utilized to improve the manuscript. The manuscript is acceptable.